# New Evidence about Skill-Biased Technological Change and Gender Wage Inequality

**Manuel Carlos Nogueira** [1,*] and **Mara Madaleno** [2]

[1] GOVCOPP—Research Unit in Governance, Competitiveness and Public Policy, ISPGAYA—Higher Polytechnic Institute of Gaya, Avenida dos Descobrimentos, 303, Santa Marinha, 4400-103 Vila Nova de Gaia, Portugal

[2] GOVCOPP—Research Unit in Governance, Competitiveness and Public Policy, Department of Economics, Management, Industrial Engineering and Tourism (DEGEIT), University of Aveiro, 3810-193 Aveiro, Portugal; maramadaleno@ua.pt

[*] Correspondence: mnogueira@ispgaya.pt

**Abstract:** In recent decades, the wage gap between higher- and lower-skilled workers has steadily widened around the world, and this gap is widening. There are several approaches in the literature to understand the causes of this steady increase, with Skill-Biased Technological Change (SBTC) being the most used and the results more consistent. This paper aims to deepen the understanding of this wage gap among workers in Organisation for Economic Cooperation and Development (OECD) countries, using cluster analysis and then modeling through simultaneous equations for the period between 2007 and 2020. Albeit with varying intensity, we conclude that in all clusters, there is a strong influence of the wage gap of the less skilled on the widening of the wage gap of the more skilled, with this influence being even more intense in the case of women. The SBTC approach can also be detected in all clusters but with greater intensity in the case of countries that invest more in research and development (R&D). Education spending and gross domestic product (GDP) per capita also play a role in widening the wage gap as well as in reducing gender inequalities. We also conclude that each cluster has its specificities that justify the choice made and that a slow reduction in gender wage inequality is observed in all clusters.

**Keywords:** skill-biased technological change; gender wage inequality; education; cluster analysis

**JEL Classification:** C30; E24; J31; O33

## 1. Introduction

Over the last fifty years, relative wages in the labor market have changed considerably, taking into account workers' qualifications, so wage differentials have continuously widened in favor of higher-skilled workers. These findings first emerged in developed countries and then spread to developing countries, and this widening wage gap became known as the skill premium (Pavcnik 2017).

Especially in the last two decades, several researchers have tried to investigate the causes and consequences of the increase in these wage differentials and to seek explanations for the fact that technological progress does not play a neutral role in the distribution of labor income. The first explanations were related to the increase in international trade between countries and the resulting diffusion of exchange, having their origin in the Stolper–Samuelson theorem (Borjas et al. 1997). Over time, however, other explanations emerged, with the SBTC approach taking precedence over the others, primarily due to the important contributions of Acemoglu (2002). For Acemoglu (2002), constant technological progress requires more and more skilled workers, but their demand increases faster than the supply of skilled workers, resulting in a constant increase in their wages relative to the wages of less skilled workers.

The increase in technology available to workers in the labor market and their better preparation at the educational level increase their productivity, and the demand for highly skilled workers continues to increase (Violante 2008; Acemoglu and Autor 2011), leading to an imbalance between the demand and supply of highly skilled labor that persists over time (Murphy and Topel 2016).

Authors such as Acemoglu and Autor (2011) and Grossman and Helpman (2018) find that the SBTC approach invisibly regulates the labor market and that as technological knowledge increases, the so-called skill premium rises as more and more highly skilled workers are needed. However, the SBTC approach poses a problem in measurement as it is not directly quantifiable. Therefore, it is necessary to use approximations of which R&D investments should be considered. Numerous authors have already studied and demonstrated the importance of R&D in the formulation of wage differentials (Machin and Van Reenen 1998; Violante 2008; Michaelsen 2011; Nogueira and Madaleno 2023) and concluded that when this type of expenditure is increased, wage differentials increase in favor of workers with higher academic qualifications.

Increasing robotization, for example, replaces medium-skilled workers with higher-skilled ones, leading to their unemployment and thus creating polarization in the labor market (Michaels et al. 2014). On one side of the pole are the best-skilled workers and on the other the least-skilled workers, so the number of jobs for middle-skilled workers gradually decreases (Damelang and Otto 2023).

The first authors to introduce the SBTC concept into the academic literature were Katz and Murphy (1992), who found for the US that the increasing wage inequality between workers based on their qualifications is due to their university degree and that this growing demand leads to imbalances in the labor market, which in turn leads to an increase in wages for graduates. Later, Wood (1998) and Acemoglu (2002) confirmed the presence of the SBTC approach in the US as well as in other Western countries such as Japan and South Korea, albeit with less intensity. According to Parnastuti et al. (2013), the differences in the intensity with which the SBTC approach is reflected in workers' wage differentials may be due to the greater or lesser speed with which highly skilled workers appear in the labor market. The SBTC approach should also be analyzed in conjunction with other variables that may affect wage rates, such as the degree of inequality, the presence of minimum wages or industry wages, efficiency wages, more or fewer barriers to hiring or firing, or investments of other kinds in the economy (Card and Lemieux 2001; Autor et al. 2008; Buera et al. 2022).

As far as we know, this is the second time that the emergence of the wage gap between more and less skilled workers has been treated as we do in this paper. The first time was in an article by Nogueira and Madaleno (2023), in which they divided OECD workers into three skill levels for the period 2007–2020: Those who completed tertiary education, those who completed high school, and those who did not complete high school, while also analyzing women's wage rates as a percentage of all workers wage rates. In this work by Nogueira and Madaleno (2023), four models were estimated, two of which estimate the wage gap between university and high school graduates and the wage gap between workers with a university degree and those with a low school degree. The other two models are similar to the first but are used to estimate the wage gap for female workers, taking into account the percentage of all worker's wage rates. These estimations were carried out considering simultaneous equations modeling. This is because the authors assumed that there are relationships between these wage differentials insofar as some workers with intermediate skills can advance to higher positions by using their work experience or vocational training. In this way, the wage gap between qualifications increases or decreases due to the reallocation of more qualified workers in terms of training. In this way, there may be mobility between these groups of workers without matching school qualifications. According to Nogueira and Madaleno (2023), this choice has proven to be correct, as wage differentials between medium-skilled workers and lower-skilled workers have been

found to have a strong impact on wage differentials between higher-skilled workers and medium-skilled workers.

Nogueira and Afonso (2018) point out that the OECD is made up of countries with very different economic and social circumstances and that wage differentials vary between most countries and different groups that can be classified according to statistical criteria. This subdivision is called cluster analysis and is what we are pursuing in this paper, as we want to explore and deepen the work of Nogueira and Madaleno (2023). Using the SBTC approach and a wide range of control variables, this article intends to understand the impact on the formation of wage differentials for workers in OECD countries, now considering group analysis.

The gap in the literature that the article intends to fill is that for the first time, through a cluster analysis considering three skill levels, workers are grouped to examine the impact of the SBTC approach in expanding their wage rates. Moreover, we studied the impact that the wage rates of medium-skilled workers relative to lower-skilled workers have on the increase in wage rates of higher-skilled workers relative to average-skilled workers. Another gap in the literature that we aim to fill is that this is the first time that this type of study has been conducted in this way for women's wage rates relative to the majority of workers. Finally, research and development, the share of unionized workers, education expenditure, globalization, GDP per capita, environmental performance, and $CO_2$ emissions per capita are the variables explored as possible explanations for these wage gaps.

The strategy followed in this paper has proven to be correct and has made it possible to draw more detailed and differentiated conclusions, implying different policy recommendations, taking into account the different groups of countries. In terms of statistical significance, a similar conclusion emerges for all clusters: In all groups, the wage gap between medium-skilled workers and less-skilled workers has a large impact on the widening of the wage gap between higher-skilled workers and less-skilled workers and average-skilled workers, albeit with different intensities depending on the cluster. This finding could imply that middle-skilled workers compete for higher wages, but that educational gains continue to reward higher skills.

The SBTC approach is present in all clusters, and this effect is felt more strongly in the countries that invest the most in R&D, as well as in the group of countries where this variable contributes the most to gender wage equalization. As for the impact of unionization, it only seems to promote the alignment of the wages of the least skilled with those of the moderately skilled by raising the wages of the least skilled. Considering education spending, the results reveal its contribution to the increase in wage inequality, but this effect is more pronounced in countries with lower GDP per capita.

Another important variable is GDP per capita. It can be seen that this variable contributes mostly to the convergence of wage differentials in the cluster consisting of the countries of Northern Europe, the US, and Australia, while it contributes to the increase in wage differentials in the other clusters analyzed.

These introductory remarks are followed by an analysis of the literature (Section 2). Section 3 then presents data, variables, statistics, and correlations. Section 4 presents the empirical analysis, Section 5 discusses the results, and Section 6 concludes the paper and presents some policy implications.

## 2. Literature Review

Although they are not the only approaches that have attempted to explain the steady increase in wage differentials between workers given their skills in recent decades, the international trade approach and the SBTC approach are the two most commonly used in the literature.

First, the contributions of the Stolper–Samuelson theorem (Borjas et al. 1997) gave rise to the international trade approach and, more recently, primarily due to Acemoglu (2002), to the SBTC approach, which eventually replaced the international trade approach.

Authors such as Lemieux (2007) and Grossman and Helpman (2018) argue that due to the ever-increasing emergence of new technologies, there is a need to hire more and more skilled workers, leading to an increase in the relative demand for these skilled workers, who will complement highly productive capital investments. The relative increase in demand for these types of workers exceeds their relative supply, raising the so-called skill premium. In a recent study conducted by Hutter and Weber (2022) for Germany, covering forty years, it was concluded that the SBTC approach leads to increases in productivity and wages, with the wage gap between skills widening and the hours worked by higher-skilled workers being reduced due to the increase in their productivity.

Recently, Taniguchi and Yamada (2022), who admit that this benefits the economy as a whole, found evidence that the skill premium between more- and less-skilled workers is increasing and that the expansion of ICT can explain this expansion.

Acemoglu and Restrepo (2021), continuing their research on the increase in the wage gap in favor of the most highly skilled, presented new approaches consisting of replacing routine tasks performed by less-skilled workers and now performed by automation, thus reducing the number of jobs required for lower skill levels, which consequently leads, on average,, to an increase in the wage gap of these workers compared to the more highly skilled and causes polarization of employment by eliminating middle-skill jobs.

Tyrowicz and Smyk (2019) also found that in developed economies, the wage gap widens in favor of higher-skilled workers when the SBTC approach is followed. Broecke et al. (2015) have previously found that wage inequality manifests and widens between the most- and least-skilled workers in both the US and other OECD countries. Similarly, looking at OECD countries, Nogueira and Afonso (2018) conclude that the SBTC approach in countries with higher GDP per capita encourages the widening of the wage gap between the most- and least-skilled workers, and as the supply of skilled workers does not keep pace with demand, the wage gap continues to increase.

Over the years, authors such as Berman et al. (1998), Kiley (1999), and Autor et al. (2008), among many others, have proposed the SBTC approach as the main reason for the increase in the wage gap between workers from different countries based on their skills, although they admit that this approach is difficult to quantify because it does not allow for direct measurement. Perhaps because it is difficult to quantify, the SBTC approach is not always tested in the studies conducted. A recent example that contradicts this approach was verified by Messina and Silva (2019), in which for South America and in a specific short-term time frame, wage inequality between graduates and non-graduates was reduced by 26% due to the sudden increase in schooling in these countries. However, other variables attempt to explain the emergence of wage differentials between workers. One of these variables relates to the unionization rate of workers. Autor et al. (2008) and Card et al. (2013) argue that the decline in union density has increased wage inequality because unions only protect the less-skilled workers and as the demand for these workers has declined, their numbers have gradually decreased, increasing wage inequality on average. Other authors reporting that the decline in union density increases wage inequality via job polarization include Biewen and Seckler (2019), who studied these effects for firms in Germany. Western and Rosenfeld (2011) take the opposite view, finding that when unions win wage increases for their members, they reduce the wage gap between these workers and higher-skilled workers, as these are mostly lower-skilled workers.

Regarding the impact of education on widening the wage gap in favor of higher-skilled workers, the opinion of authors is unanimous in both older and more recent studies, as the vast majority find evidence that an increase in education spending causes the widening of this wage gap (Benabou 2000; Muinelo-Gallo and Roca-Sagalés 2011; Antonczyk et al. 2018; Nogueira and Afonso 2018; Jacobs and Thuemmel 2022). These authors believe that if a country increases its investment in education, its students will advance in their schooling and they will receive higher wages than the less qualified when they enter the labor market, as they will be more productive, thus widening the wage gap.

Regarding the impact of GDP per capita on wage inequality, Kuznets (1955) found that an increase in this indicator should be accompanied by a reduction in wage inequality, as economic growth allows people to invest more in education, leading to a supply of skilled workers with higher education, which reduces the wage gap with less-skilled workers. Nogueira and Afonso (2018) also reached similar conclusions. The introduction of this variable in our study is justified by the fact that OECD countries have relatively large differences in GDP per capita among themselves.

Globalization can cause narrow or widening effects on the wage gap (Oostendorp 2009). The only study published to date that uses the Globalization Economic Index as a possible explanation for the formulation of wage differentials between workers was by Nogueira and Madaleno (2023), who conclude that it does not contribute significantly to exacerbating or reducing wage differentials. Jestl et al. (2022), using certain variables related to globalization, conclude that immigration is responsible for the increase in wage inequality at the middle and top of the wage distribution, which may mean that immigrants are willing to accept lower wages than natives for the same type of work. Previously, Meschi et al. (2016) reported that globalization increases the wage gap between more- and less-skilled workers. Furthermore, globalization and research and development (leading to innovation) promote economic development, pushing the development process of industrialization and urbanization and increasing GDP per capita (Wang et al. 2023). Thus, if innovation is present, it requires highly skilled workers to manage these new technologies, only achieved through more education or formation. However, innovation and new technologies also lead to lower production costs and higher productivity, stimulating higher GDP or economic growth. If, in certain regions, the differences in economic growth between countries are due to the technologies used, which play an important role in increasing productivity and reducing costs, it becomes important to examine the evolution of technological wages compared to non-technological wages (Çaliskan 2015). Wahiba and Mahmoudi (2023) further state that globalization and skill-based technological workers emerged as justifications for further inequalities and wage gap rise. This wage gap increase may be justified by the fact that the adoption of new technologies requires a high level of human capital, which is often scarce at the beginning of the process of technological diffusion. Hence, improving human capital is an essential element for economic growth and will reduce inequality (Topuz 2022). Additionally, more competent and educated people have assisted in abatement technology invention and dissemination.

Several authors point out that in the short term, there is a trade-off between economic growth and environmental protection since boosting production and consumption activities leads to environmentally harmful emissions. To protect the environment, production and consumption activities would have to be scaled back, which could affect economic growth (Tang et al. 2019; Marsiglio and Privileggi 2021). In the long run, however, authors' views are already changing somewhat, as they argue that environmental concerns are compatible with economic growth so that the costs of environmental growth are limited to short-run effects. These authors include Porter (1991), Bashir et al. (2021), and Marsiglio and Privileggi (2021), who add that firms rationalize their operations and develop greener technologies to return to sustainable economic growth. Nogueira and Madaleno (2021) also take a similar view, demonstrating that it is possible to achieve economic growth without neglecting the environment and the sustainability of the planet. Krueger et al. (2021) found growing evidence that more skilled workers are willing to take a pay cut to work in companies that operate in sectors that are concerned about environmental sustainability. Bunderson and Thakor (2020) also reach the same conclusions, arguing that wage cuts are only accepted by better-qualified workers. Firms with greater human capital tend to be more long-run focused, which promotes sustainable growth (Khan et al. 2022). As a result, industrial enterprises tend to have stricter environmental controls and larger human capital reserves. They are also less likely to infringe on external environmental standards (Li and Ullah 2022). Because globalization raises wealth and output, it is eventually aided

by technology spillovers in exporting economies. After a given period, globalization may reduce energy consumption and carbon dioxide emissions (Ahmed et al. 2021).

Although wage gaps between men and women still exist, several authors argue that these gaps have weakened in recent decades as women acquire a greater number of skills and compete with men for higher wages and more demanding, well-paid jobs (Shen 2014; Kovalenko and Töpfer 2021). Moore (2018) also adds that we are facing a process of pay convergence between men and women.

On the other hand, Damelang and Otto (2023) argue that robotization contributes to a growing skills shortage and increasing occupational inequality, which is reflected in employment and wage levels. Nevertheless, middle-skilled workers are the most affected by robotization, as the most skilled have better adaptability due to their greater share of skills. In addition, Picatoste et al. (2023) conduct a cluster analysis classifying countries according to their gender digital divide concluding the educational level impact.

## 3. Data, Variables, Statistics, and Correlations

The sample we use in our empirical analysis includes only 34 OECD countries for the period between 2007 and 2020. For countries such as Iceland, Costa Rica, Lithuania, and Colombia, a great deal of data are missing, so they cannot be used. For the 34 countries considered, some information is missing, so the whole database can be considered only 396 observations for all workers and 333 for women instead of 476 observations.

Table 1 shows the variables, units of measurement, and data source used in the empirical analysis and Table 2 shows the main descriptive statistics and correlations for the whole sample. In Table 2, the variables WGH, WGM, and WGL represent the wage rates of workers who have tertiary education, are high school graduates, and are below high school graduates, respectively. In addition, the variables WGWH, WGWM, and WGWL represent the salary rates of women who have tertiary education, are high school graduates, and are below high school graduates, respectively, but as a percentage of the wage rates of all workers. To obtain accurate results from the empirical analysis, we also consider the problem of multicollinearity. When applied to our variables, the Pearson correlation test (Table 2) showed that there is no multicollinearity between the variables considered. Following Madanipour and Thompson (2020), we used the value of 0.80 as the cut-off value as postulated by some renowned econometricians, even though there is no absolute consensus on this value. Table 3 shows the average values for each variable and country considered in our study.

Since the presence of multicollinearity is of great importance, we decided to perform the de facto calculation of variance inflation (VIF). According to Craney and Surles (2002), multicollinearity exists between at least two of the regressors if the independent variables are not orthogonal. In this case, the associated parameters can quickly lose the explanatory power of their variables. For the same authors, the variance inflation factor (VIF) is one of the statistical tests used to measure the degree of multicollinearity for each of the variables. If it exceeds the value of 10, this is often seen as an indicator that multicollinearity is overly affecting the estimates. As we can see from the results in Table 4, no VIF value exceeds the reference value for the cutoff, so we assume that multicollinearity is reduced.

**Table 1.** Variable definition and data source.

| Variable | Definition | Unit | Source |
|---|---|---|---|
| $WGH_{i,t}/WGM_{i,t}$ | Gap between wage rates of university graduates and high school graduates in country i and year t, in real terms | Index | OECD Education at a Glance—Kovalenko and Töpfer (2021); Acemoglu and Restrepo (2018, 2021) |
| $WGM_{i,t}/WGL_{i,t}$ | Gap between wage rates of high school graduates and bellow high school graduates in country i and year t, in real terms. | Index | OECD Education at a Glance—Kovalenko and Töpfer (2021); Acemoglu and Restrepo (2018, 2021) |
| $WGWH_{i,t}/WGWM_{i,t}$ | Gap between women's wage rates of university graduates and high school graduates in country i and year t, in real terms, as a percentage of the wage rates of all workers. | Index | OECD Education at a Glance—Kovalenko and Töpfer (2021); Acemoglu and Restrepo (2018, 2021) |
| $WGWM_{i,t}/WGWL_{i,t}$ | Gap between women's wage rates of high school graduates and below high school in country i and year t, in real terms, as a percentage of the wage rates of all workers. | Index | OECD Education at a Glance—Kovalenko and Töpfer (2021); Acemoglu and Restrepo (2018, 2021) |
| $SBTC_{i,t}$ | Research and Development spending as a percentage of GDP in country i and year t | Percentage | OECD—Acemoglu and Restrepo (2018, 2021); Kristal and Cohen (2017) |
| $Union_{i,t}$ | Share of unionized workers in country i and year t | Percentage | OECD—Kristal and Cohen (2017) |
| $EPI_{i,t}$ | Environmental Performance Index, in the country i and year t | Index | Environmental Law & Policy—Hsu and Zomer (2016); Wendling et al. (2018) |
| $Educ.Expend_{i,t}$ | Education expenditure as a percentage of GDP in country i and year t | Percentage | OECD Education at a Glance—Nogueira and Afonso (2018) |
| $CO_2$ | $CO_2$ emissions per capita in country i and year t | Tons | World Bank—Nogueira and Madaleno (2021) |
| $KOF_{i,t}$ | Globalization Economic Index in country i and year t | Index | KOF Swiss Economic Institute |
| $GDP\ pc_{i,t}$ | Gross domestic product per capita in country i and year t, US dollar constant prices, 2015 PPPs | Value in dollars | OECD World Bank—Nogueira and Afonso (2018) |

Source: Authors' elaboration.

**Table 2.** Main descriptive statistics and correlations.

| | WGH | WGM | WGL | WGWH | WGWM | WGWL | SBTC | Union | EPI | Educ. Expend. | CO$_2$ | KOF | GDPpc | Average | Standard Deviation | Max | Min |
|---|---|---|---|---|---|---|---|---|---|---|---|---|---|---|---|---|---|
| WGH | - | 0.06 | −0.48 | −0.13 | 0.11 | 0.02 | −0.36 | −0.41 | −0.24 | −0.29 | −0.23 | −0.25 | −0.40 | 154.63 | 23.361 | 260 | 115 |
| WGM | | - | −0.18 | 0.09 | 0.06 | 0.10 | 0.18 | 0.16 | 0.21 | 0.04 | 0.11 | 0.18 | 0.29 | 107.72 | 12.360 | 146 | 61 |
| WGL | | | - | 0.11 | 0.03 | 0.06 | 0.18 | 0.37 | 0.23 | 0.12 | 0.11 | 0.33 | 0.15 | 78.221 | 8.1625 | 101 | 54 |
| WGWH | | | | - | 0.38 | 0.23 | −0.03 | 0.23 | 0.16 | −0.02 | −0.06 | 0.07 | 0.14 | 75.525 | 7.1177 | 148 | 61 |
| WGWM | | | | | - | 0.64 | 0.08 | 0.25 | 0.08 | −0.21 | −0.32 | 0.26 | 0.13 | 77.080 | 6.6743 | 98 | 54 |
| WGWL | | | | | | - | 0.13 | 0.44 | 0.19 | −0.07 | −0.15 | 0.40 | 0.38 | 76.154 | 6.6814 | 92 | 49 |
| SBTC | | | | | | | - | 0.41 | 0.18 | 0.29 | 0.18 | 0.35 | 0.36 | 1.9327 | 1.0352 | 4.93 | 0.28 |
| Union | | | | | | | | - | 0.39 | 0.35 | 0.06 | 0.47 | 0.46 | 24.813 | 17.418 | 72.5 | 4.53 |
| EPI | | | | | | | | | - | 0.22 | 0.22 | 0.44 | 0.41 | 79.770 | 8.3420 | 90.8 | 42.6 |
| Educ. Expend. | | | | | | | | | | - | 0.05 | −0.13 | 0.07 | 5.4694 | 1.0424 | 8.42 | 3.25 |
| CO$_2$ | | | | | | | | | | | - | 0.13 | 0.45 | 8.6885 | 4.0938 | 23.8 | 2.77 |
| KOF | | | | | | | | | | | | - | 0.54 | 82.021 | 5.8417 | 90.9 | 61.8 |
| GDPpc | | | | | | | | | | | | | - | 38,045 | 23,153 | 116,597 | 8002 |

Source: Authors' elaboration.

**Table 3.** Average of variables for each OECD country (2007–2020). Source: Authors' elaboration.

| Country | WGH | WGM | WGL | WGWH | WGWM | WGWL | SBTC (%) | Union (%) | EPI | Educ. Exp. | CO$_2$ | KOF | GDPpc |
|---|---|---|---|---|---|---|---|---|---|---|---|---|---|
| Australia | 132.85 | 97.35 | 83.5 | 77.72 | 75.81 | 80.01 | 2.01 | 16.26 | 83.78 | 5.67 | 17.42 | 80.54 | 55,856 |
| Austrium | 153.57 | 118.28 | 69.14 | 73.54 | 79.54 | 76.81 | 2.93 | 28.09 | 83.07 | 5.28 | 7.92 | 86.94 | 44,460 |
| Belgium | 133.71 | 99.35 | 88.78 | 81.36 | 82.63 | 80.82 | 2.54 | 52.46 | 77.80 | 6.11 | 9.31 | 89.51 | 40,622 |
| Canada | 140.42 | 113.71 | 81.71 | 72.63 | 70.54 | 66.81 | 1.75 | 26.76 | 81.22 | 6.21 | 16.08 | 82.87 | 42,771 |
| Chile | 246.51 | - | 67.25 | 66.25 | 72.00 | 78.00 | 0.36 | 14.68 | 72.12 | 6.28 | 4.41 | 76.33 | 12,826 |
| Czech Republic | 175.92 | - | 72.57 | 71.63 | 79.83 | 79.91 | 1.67 | 13.92 | 79.53 | 4.32 | 10.37 | 83.09 | 17,674 |
| Denmark | 127.14 | 102.57 | 82.42 | 77.01 | 80.54 | 81.91 | 2.93 | 68.20 | 86.55 | 7.09 | 7.19 | 87.72 | 53,587 |
| Estonia | 132.63 | 89.66 | 89.91 | 70.27 | 61.54 | 61.18 | 1.53 | 6.12 | 81.01 | 5.21 | 12.92 | 80.93 | 17,320 |
| Finland | 143.85 | 119.14 | 95.35 | 77.54 | 78.18 | 79.72 | 3.17 | 67.22 | 87.25 | 5.91 | 9.43 | 86.58 | 44,329 |
| France | 149.50 | 89.66 | 83.42 | 74.45 | 80.18 | 74.63 | 2.19 | 10.78 | 85.27 | 5.70 | 5.42 | 86.50 | 36,620 |
| Germany | 164.28 | 112.01 | 82.78 | 73.90 | 82.27 | 76.63 | 2.88 | 17.93 | 81.68 | 4.65 | 9.67 | 87.37 | 40,276 |
| Greece | 146.91 | 102.09 | 75.27 | 74.72 | 78.54 | 68.82 | 0.88 | 21.72 | 79.85 | 3.71 | 7.85 | 80.32 | 19,654 |
| Hungary | 204.01 | 109.92 | 74.42 | 72.91 | 87.72 | 83.18 | 1.24 | 11.06 | 76.38 | 4.50 | 5.03 | 84.27 | 12,575 |
| Ireland | 167.42 | 96.92 | 85.71 | 75.36 | 77.15 | 80.45 | 1.37 | 27.97 | 84.13 | 4.94 | 8.61 | 85.46 | 56,989 |
| Israel | 154.02 | 111.87 | 76.28 | 69.90 | 75.27 | 72.72 | 4.39 | 26.03 | 75.98 | 6.51 | 8.25 | 76.82 | 35,040 |
| Italy | 147.85 | - | 78.14 | 72.90 | 76.72 | 77.45 | 1.30 | 34.05 | 80.62 | 4.35 | 6.49 | 81.51 | 34,981 |
| Japan | 150.27 | - | 78.72 | - | - | - | 3.21 | 17.82 | 78.85 | 4.58 | 9.46 | 75.07 | 40,898 |
| Korea | 143.21 | - | 70.85 | 67.36 | 65.18 | 66.72 | 3.78 | 10.29 | 69.02 | 6.66 | 12.17 | 75.82 | 27,218 |
| Latvia | 145.20 | 98.40 | 88.80 | 77.40 | 71.80 | 69.60 | 0.59 | 13.49 | 78.76 | 4.42 | 4.72 | 75.03 | 14,981 |
| Luxembourg | 153.28 | 125.12 | 71.71 | 79.45 | 79.45 | 81.72 | 1.35 | 34.39 | 84.66 | 3.73 | 19.44 | 85.48 | 110,257 |
| Mexico | 192.33 | 120.35 | 62.16 | 69.16 | 77.40 | 72.66 | 0.40 | 13.90 | 67.92 | 5.59 | 3.93 | 67.07 | 9618 |
| Netherlands | 152.57 | 114.35 | 83.35 | 77.36 | 81.27 | 81.09 | 1.97 | 18.29 | 79.25 | 5.55 | 9.73 | 89.07 | 51,446 |
| New Zealand | 127.71 | 110.07 | 83.64 | 77.45 | 77.27 | 79.27 | 1.24 | 19.42 | 84.02 | 6.64 | 8.59 | 76.71 | 38,626 |
| Norway | 126.38 | 114.85 | 79.07 | 75.36 | 77.37 | 80.82 | 1.82 | 50.03 | 84.42 | 6.87 | 9.75 | 84.81 | 85,543 |
| Poland | 167.50 | 104.57 | 83.28 | 76.81 | 77.45 | 71.72 | 0.87 | 15.53 | 68.29 | 5.18 | 8.51 | 78.77 | 12,205 |
| Portugal | 166.35 | 101.14 | 70.14 | 73.54 | 74.00 | 71.74 | 1.37 | 17.60 | 74.39 | 5.53 | 4.98 | 82.39 | 19,728 |
| Slovak Republic | 171.72 | 131.47 | 68.18 | 70.27 | 73.91 | 73.36 | 0.75 | 14.05 | 79.11 | 4.12 | 6.67 | 81.48 | 17,781 |
| Slovenia | 181.71 | - | 76.85 | 86.09 | 86.18 | 84.36 | 2.07 | 30.69 | 81.02 | 5.03 | 7.38 | 79.32 | 24,177 |
| Spain | 142.64 | 109.0 | 79.35 | 84.18 | 76.58 | 76.08 | 1.27 | 15.85 | 84.29 | 4.71 | 5.97 | 83.59 | 29,731 |
| Sweden | 123.78 | 114.85 | 83.28 | 81.18 | 81.81 | 85.03 | 3.28 | 67.40 | 86.32 | 6.02 | 4.57 | 88.78 | 54,692 |
| Switzerland | 153.50 | 109.12 | 76.35 | 78.66 | 83.83 | 78.83 | 3.11 | 16.36 | 85.64 | 5.19 | 4.98 | 89.34 | 82,481 |
| Turkey | 160.92 | - | 69.57 | 83.57 | 80.43 | 69.00 | 0.86 | 7.75 | 59.38 | 4.60 | 4.71 | 68.36 | 10,583 |
| United Kingdom | 154.53 | - | 71.21 | 76.63 | 72.36 | 74.90 | 1.65 | 25.43 | 85.66 | 6.14 | 6.92 | 88.58 | 43,096 |
| United States | 174.46 | 108.5 | 67.64 | 69.90 | 71.00 | 70.72 | 2.82 | 10.84 | 79.62 | 6.72 | 17.22 | 81.13 | 54,888 |

**Table 4.** Variance inflation factor (VIF).

| | |
|---|---|
| SBTC | 1.353 |
| Union | 1.624 |
| EPI | 1.353 |
| Educ. Expend | 1.267 |
| CO$_2$ | 1.391 |
| KOF | 1.723 |
| GDPpc | 1.921 |

Source: Authors' elaboration.

## 4. Empirical Analysis, Model Specification, and Estimation Methods

As mentioned above, this paper aims to elaborate on the paper developed by Nogueira and Madaleno (2023) in an original way (as well as to consider suggestions for further work) and to test the impact of the SBTC approach and a set of control variables in the formation of the wage differential between workers in OECD countries who have secondary education compared to those who do not on the wages of those with higher education compared to those with secondary education. The work is deepened by a cluster analysis of OECD countries and subsequent estimations by simultaneous equations. We have also carried out the same approaches for the case of women, with wage rates as a percentage of all workers.

*Exploratory Multivariate Analysis Technique*

Interest in the study of agglomerations began with the economist Alfred Marshall, who devoted a chapter to the external effects of specialized industrial locations in his book "Principles of Economics" (1890).

Cluster analysis, also called taxonomy analysis and segmentation analysis, was first used by Tyron (1939) and consists of a variety of different classification algorithms, all dealing with the question of how to group observed data into meaningful structures or how to develop taxonomies capable of classifying observed data into different groups. However, the word cluster was first used by Porter (1990). In this book, the author demonstrated that clusters occur in several cities and regions, in different sectors and types of technology, and that they are often the main source of comparative advantage for many countries when compared internationally. Since then, many countries have promoted this type of clustering to support the most promising sectors.

A common problem is to ensure, given a set of n observations, that these observations are grouped into classes in such a way that each of these classes is internally homogeneous (i.e., consists of observations that are considered "similar") and that the different classes are heterogeneous among themselves (i.e., the observations of the different classes are "dissimilar"). Discriminant analysis methods assume that such a division is known in an available dataset, and the goal is to look for directions in space that show the separation of these subgroups or to determine a rule for future classifications. In many cases, however, there is no classification, and the problem is to determine which (and how many) different classes of observations exist in the available data. Methods that allow such classes to be determined are called classification analysis methods.

For Everitt (1993), cluster analysis is a statistical technique that aims to group observations that share similar characteristics based on a set of variables in the sample. Cluster analysis thus classifies observations into homogeneous groups called clusters or conglomerates. It is thus assumed that the groups formed by cluster analysis are similar to each other (within the cluster, the variance is minimal) and different from other clusters (between the clusters the variance is maximal). To perform the cluster analysis, we need to consider the average values of the observations available for each country for each of the variables, as shown in Table 5.

**Table 5.** Average of variables for each cluster.

| Cluster | WGH | WGM | WGL | WGWH | WGWM | WGWL | SBTC (%) | Union (%) | EPI | Educ. Exp. | CO$_2$ | KOF | GDPpc |
|---------|-----|-----|-----|------|------|------|----------|-----------|-----|-----------|--------|-----|-------|
| Cluster 1 | 140.60 | 112.77 | 77.71 | 76.52 | 79.58 | 80.11 | 1.514 | 21.51 | 85.03 | 6.031 | 5.955 | 87.07 | 84,012 |
| Cluster 2 | 146.38 | 105.18 | 81.17 | 76.42 | 77.90 | 79.87 | 1.504 | 22.84 | 83.27 | 6.001 | 10.26 | 85.44 | 54,576 |
| Cluster 3 | 153.28 | 125.12 | 71.71 | 79.45 | 78.27 | 81.72 | 1.981 | 18.29 | 84.66 | 3.714 | 9.730 | 85.48 | 110,257 |
| Cluster 4 | 146.40 | 110.11 | 79.97 | 75.14 | 76.38 | 75.22 | 2.141 | 19.85 | 81.07 | 5.634 | 8.314 | 82.99 | 38.359 |
| Cluster 5 | 173.59 | 149.75 | 74.83 | 74.90 | 76.91 | 73.92 | 1.823 | 32.09 | 75.31 | 4.978 | 8.720 | 79.00 | 15,760 |

Source: Authors' elaboration.

Since the variables are not expressed in the same unit of measurement and to make an apportionment based on the homogenization of the units of measurement, it is necessary to normalize the data (in terms of mean and standard deviation) for each OECD country (Garson 2008).

The hierarchical class aggregation method that we are going to use is Ward's (1963) method, which is the most followed, as it is considered the most robust (Murtagh and Legendre 2014; Majerova and Nevima 2017). Although this method is sixty years old, it has lost none of its validity. It tries to find the greatest homogeneity of individuals belonging to a cluster and to minimize the variance concerning the centers, which allows for minimizing the sum of squared errors (Vita et al. 2021). For this reason, we use the Euclidean distance squared as a measure of the distance between countries (Strauss and Von Maltitz 2017).

The first step we will take is a hierarchical analysis, in which we create the dendrogram (Figure 1) using Ward's method and, as mentioned earlier, the squared Euclidean distance. From this analysis, we can determine that the ideal solution for the number of clusters is five. We will then use the non-hierarchical method to test whether the hypothesis of four clusters is confirmed or whether it would be better to divide the countries into five clusters, i.e., which of the two solutions has better homogeneity.

In the non-hierarchical method, the composition of the number of countries in each cluster is shown in Table 6.

**Table 6.** Number of countries per cluster.

|  | Number of Countries | Number of Countries |
|---|---|---|
| Cluster 1 | 1 | 2 |
| Cluster 2 | 17 | 6 |
| Cluster 3 | 2 | 1 |
| Cluster 4 | 14 | 13 |
| Cluster 5 |  | 12 |
| Valid | 34 | 34 |
| Missing | 0 | 0 |

Source: Authors' elaboration.

As we can see in Table 6, for both cases where the ideal number of clusters is four or five, there is a cluster with only one country (which is considered an outlier) and another cluster with only two countries. In both the one and the other case, Luxembourg stands alone, and the two countries are Norway and Switzerland, which proves the great heterogeneity between these countries compared to the others.

To support the decision on the best solution for the number of clusters to consider, we will resort to the Criterion of Caliński and Harabasz (1974). Often called the variance ratio criterion, it assumes the following expression:

$$VRCk = SSB/SSW \times (N-K)/(K-1) \tag{1}$$

$SSB$ is the square of the sum of the distance between clusters, $SSW$ is the square of the sum of the total distance within each cluster, $N$ is the number of observations, and $K$ is the number of clusters.

If the choice of the number of clusters is correct, they will have a large distance between them ($SSB$) and a small variation within the cluster ($SSW$) with respect to the centroid. The higher the value in the $VRCk$ statistic, the better the data partition. To determine the optimal number of clusters, we need to maximize $VRCk$ with respect to k.

According to Caliński and Harabasz (1974), the ideal number of clusters is the one with the highest index value. In our case, VRC4 = 11,328 and VRC5 = 18,751, so according to Caliński and Harabasz (1974), the ideal number of clusters is five. The division into five clusters provides better homogeneity within each group and better heterogeneity between clusters.

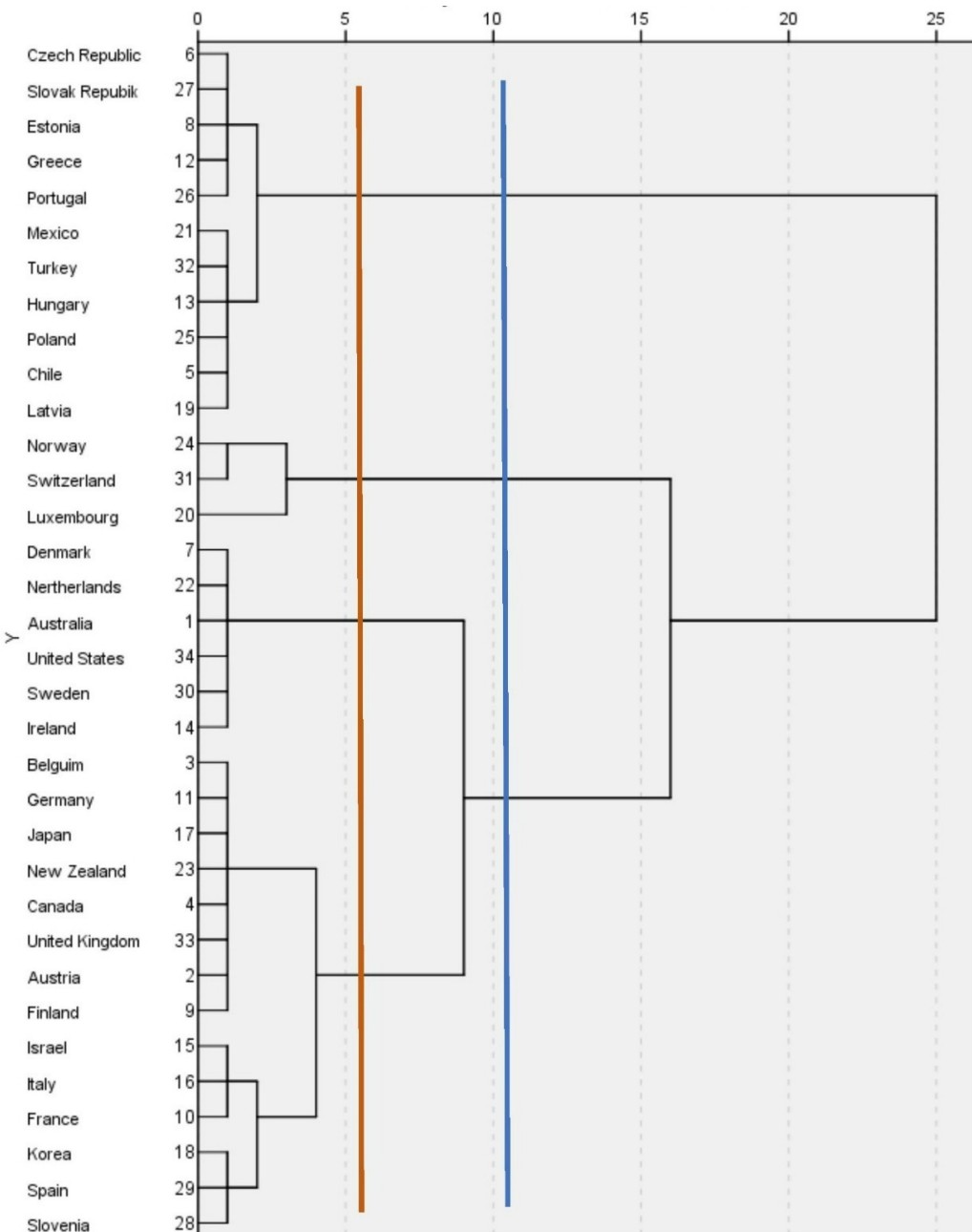

**Figure 1.** Dendrogram using the Ward Linkage. Source: Authors' elaboration.

From the analysis of Table 5, which shows the average of each variable in each cluster and the composition of each cluster, we can conclude the following.

-   Cluster 1 consists only of Norway and Switzerland. The three most prominent characteristics of this cluster are the high environmental performance index, the highest expenditure on education as a percentage of GDP, and the lowest average $CO_2$ emissions per capita.
-   Cluster 2 includes Australia, Denmark, Ireland, the Netherlands, Sweden, and the United States. This cluster has the highest $CO_2$ emissions per capita and the lowest wage gap for all education levels, between most workers and women.
-   Cluster 3 consists only of Luxembourg. This country has the highest GDP per capita and the lowest education expenditure as a percentage of GDP.

- Cluster 4 includes Austria, Belgium, Canada, Finland, France, Germany, Israel, Italy, Japan, Korea, New Zealand, Spain, and the United Kingdom. This cluster is the one with the highest average R&D as a percentage of GDP.
- Cluster 5 consists of Chile, the Czech Republic, Estonia, Greece, Hungary, Latvia, Mexico, Poland, Portugal, the Slovak Republic, Slovenia, and Turkey. This group of countries has the lowest GDP per capita, the lowest environmental performance index, and the highest percentage of trade union membership.

As mentioned above and taking into account the clustering we have conducted, we intend to examine the influence of seven variables in forming the wage gap between workers in OECD countries who have completed tertiary education and those who have completed only secondary education, as well as those who have completed only high school and those who have not completed high school. We also intend to test for women only (as a percentage of men's salary index) regarding the same effects of these seven variables, which are shown in Table 1.

Since cluster 1 includes only two countries (Norway and Switzerland) and cluster 3 includes only Luxembourg, there are not enough observations to obtain panel estimators, so we only make estimates for the remaining three clusters.

Since there is evidence of simultaneity relationships between university graduates and high school graduates, and between the latter and those who did not complete school, the econometric estimations were carried out using a system of simultaneous equations, as already performed by Nogueira and Madaleno (2023). The structural form of the equations is as follows:

$$\frac{\text{LnWGH}_{i,t}}{\text{LnWGM}_{i,t}} = \alpha_i + \beta_1 \frac{\text{LnWGM}_{i,t}}{\text{LnWGL}_{i,t}} + \beta_2 \text{LnSBTC}_{i,t} + \beta_3 \text{LnUnion}_{i,t} + \beta_4 \text{LnEPI}_{i,t} + \beta_5 \text{LnEduc.expend.}_{i,t} + \beta_6 \text{LnCO2}_{i,t} + \beta_7 \text{LnGDPpc}_{i,t} + \mu_{i,t} \tag{2}$$

$$\frac{\text{LnWGM}_{i,t}}{\text{LnWGL}_{i,t}} = \alpha_i + \sigma_1 \text{LnSBTC}_{i,t} + \sigma_2 \text{LnUnion}_{i,t} + \sigma_3 \text{LnEPI}_{i,t} + \sigma_4 \text{LnEduc.Expend.}_{i,t} + \sigma_5 \text{LnCO2}_{i,t} + \sigma_6 \text{LnKOF}_{i,t} + \sigma_7 \text{LnGDPpc}_{i,t} + \mu_{i,t} \tag{3}$$

$$\frac{\text{LnWGWH}_{i,t}}{\text{LnWGWM}_{i,t}} = \alpha_i + \beta_1 \frac{\text{LnWGWM}_{i,t}}{\text{LnWGWL}_{i,t}} + \beta_2 \text{LnSBTC}_{i,t} + \beta_3 \text{LnUnion}_{i,t} + \beta_4 \text{LnEPI}_{i,t} + \beta_5 \text{LnEduc.Expend.}_{i,t} + \beta_6 \text{LnCO2}_{i,t} + \beta_7 \text{LnGDPpc}_{i,t} + \mu_{i,t} \tag{4}$$

$$\frac{\text{LnWGWM}_{i,t}}{\text{LnWGWL}_{i,t}} = \alpha_i + \sigma_1 \text{LnSBTC}_{i,t} + \sigma_2 \text{LnUnion}_{i,t} + \sigma_3 \text{LnEPI}_{i,t} + \sigma_4 \text{LnEduc.Espend.}_{i,t} + \sigma_5 \text{LnCO2}_{i,t} + \sigma_6 \text{LnKOF}_{i,t} + \sigma_7 \text{LnGDPpc}_{i,t} + \mu_{i,t} \tag{5}$$

Equation (2) regresses the wage differential between workers who have completed tertiary education and those who have only completed high school. However, since simultaneity relationships are suspected, it includes the wage gap between workers who have completed secondary education and workers who have not completed secondary education in addition to six of the seven independent variables identified in Table 1. Equation (3), in turn, regresses the wage gap between workers who have completed secondary education and workers who have not and also includes six of the seven independent variables identified in Table 1. Equations (4) and (5), in turn, include the same variables as Equations (2) and (3), but refer only to female workers as a percentage of the wage rates for male workers.

As mentioned by Nogueira and Madaleno (2023), modeling with simultaneous equations allows for the analysis not only of the individual behavior of each equation but also of the possible relationships between equations and variables in a given period. Through this methodology, it becomes possible to increase the accuracy of the model estimates by using additional information from the correlations that support more reliable measurements.

For the identification of the structural equations of the system of simultaneous equations, the condition of order was considered because, according to Gujarati and Porter (2008), this condition is sufficient in practice to ensure identifiability when the number of equations is only two. As we see in Table 7, the four structural equations of the system can be considered exactly identified. Therefore, we can use the two-stage least squares (2SLS)

method to estimate the structural parameters, which can solve the potential problem of endogeneity (Gujarati and Porter 2008).

**Table 7.** Identification by order of the simultaneous equations model.

| Equation Number | K − k | m − 1 | K − k ≥ m − 1 | Identification |
|---|---|---|---|---|
| (1) | 9 − 8 | 1 | 1 ≥ 1 | Exactly identified |
| (2) | 9 − 8 | 1 | 1 ≥ 1 | Exactly identified |
| (3) | 9 − 8 | 1 | 1 ≥ 1 | Exactly identified |
| (4) | 9 − 8 | 1 | 1 ≥ 1 | Exactly identified |

Source: Authors' elaboration.

It should be noted that authors such as Henningsen and Hamann (2007) argue that while the estimators obtained with the 2SLS method are consistent, estimation with the three-stage least squares (3SLS) method asymptotically yields even more efficient estimators, which is why we will use this method to determine the structural parameters. Complementing this, Henningsen and Hamann (2007) argue that the best estimation efficiency is obtained with the 3SLS method when using the matrix of estimated least squares moments of two levels of structural perturbations to estimate the coefficients of the whole system simultaneously. The results obtained are shown in Tables 8–10 and are discussed in the next section. Table 11 presents the main statistics for clusters 2, 4, and 5.

**Table 8.** Three-stage least squares regression—cluster 2.

| Variables | Coefficients | Variables | Coefficients |
|---|---|---|---|
| LnWGH/LnWGM | | LnWGWH/LnWGWM | |
| LnWGM/LnWGL | 0.28753 *** | LnWGWM/LnWGWL | 0.29931 *** |
| LnSBTC | 0.08913 ** | LnSBTC | 0.09153 ** |
| LnUnion | 0.00154 | LnUnion | 0.00348 |
| LnEPI | −0.07512 | LnEPI | 0.03283 |
| LnEduc.Expend. | 0.1054 *** | LnEduc.Expend. | 0.12325 *** |
| $LnCO_2$ | 0.03714 ** | $LnCO_2$ | 0.03118 * |
| LnGDPpc | 0.01573 ** | LnGDPpc | 0.01731 *** |
| Constant | 1.18325 *** | Constant | 1.32751 *** |
| LnWGM/LnWGL | | LnWGWM/LnWGWL | |
| LnSBTC | 0.07143 *** | LnSBTC | 0.08315 *** |
| LnUnion | −0.00254 ** | LnUnion | 0.00208 * |
| LnEPI | −0.10322 | LnEPI | −0.08371 |
| LnEduc.Expend. | 0.09163 ** | LnEduc.Expend. | 0.10351 * |
| $LnCO_2$ | 0.01573 | $LnCO_2$ | 0.02167 * |
| LnKOF | 0.06811 ** | LnKOF | 0.07363 ** |
| LnGDPpc | 0.01352 ** | LnGDPpc | 0.0184 ** |
| Constant | 0.95321 *** | Constant | 0.83151 *** |

Note: ***, **, and * denote statistical significance at the 1%, 5%, and 10% levels of significance, respectively. Source: Authors' calculations.

**Table 9.** Three-stage least squares regression—cluster 4.

| Variables | Coefficients | Variables | Coefficients |
|---|---|---|---|
| LnWGH/LnWGM | | LnWGWH/LnWGWM | |
| LnWGM/LnWGL | 0.31728 *** | LnWGWM/LnWGWL | 0.34176 *** |
| LnSBTC | 0.14325 *** | LnSBTC | 0.16032 *** |
| LnUnion | 0.09325 | LnUnion | 0.11327 |
| LnEPI | 0.00477 * | LnEPI | 0.00328 |
| LnEduc.Expend. | 0.10743 ** | LnEduc.Expend. | 0.11128 *** |
| $LnCO_2$ | 0.01871 * | $LnCO_2$ | 0.02174 * |
| LnGDPpc | 0.06122 ** | LnGDPpc | 0.07312 *** |
| Constant | 1.38751 *** | Constant | 1.07112 *** |
| LnWGM/LnWGL | | LnWGWM/LnWGWL | |
| LnSBTC | 0.06740 *** | LnSBTC | 0.07312 *** |
| LnUnion | 0.08212 | LnUnion | 0.09157 * |
| LnEPI | 0.11723 | LnEPI | 0.10124 |
| LnEduc.Expend. | 0.06188 *** | LnEduc.Expend. | 0.07774 *** |
| $LnCO_2$ | 0.00216 ** | $LnCO_2$ | 0.02767 ** |
| LnKOF | 0.03754 | LnKOF | 0.03284 |
| LnGDPpc | 0.02774 *** | LnGDPpc | 0.03861 ** |
| Constant | 0.83114 *** | Constant | 0.91718 *** |

Note: ***, **, and * denote statistical significance at the 1%, 5%, and 10% levels of significance, respectively. Source: Authors' calculations.

**Table 10.** Three-stage least squares regression—cluster 5.

| Variables | Coefficients | Variables | Coefficients |
|---|---|---|---|
| LnWGH/LnWGM | | LnWGWH/LnWGWM | |
| LnWGM/LnWGL | 0.41126 *** | LnWGWM/LnWGWL | 0.38651 *** |
| LnSBTC | 0.05138 ** | LnSBTC | 0.06274 *** |
| LnUnion | 0.03251 | LnUnion | 0.04715 |
| LnEPI | 0.07680 | LnEPI | 0.05317 |
| LnEduc.Expend. | 0.15712 *** | LnEduc.Expend. | 0.16321 ** |
| $LnCO_2$ | 0.03711 ** | $LnCO_2$ | 0.03255 *** |
| LnGDPpc | 0.05893 *** | LnGDPpc | 0.06925 *** |
| Constant | 0.79883 *** | Constant | 1.12630 *** |
| LnWGM/LnWGL | | LnWGWM/LnWGWL | |
| LnSBTC | 0.06122 *** | LnSBTC | 0.07312 *** |
| LnUnion | 0.10377 | LnUnion | 0.1186 *** |
| LnEPI | 0.02614 | LnEPI | 0.03744 |
| LnEduc.Expend. | 0.09683 *** | LnEduc.Expend. | 0.10328 *** |
| $LnCO_2$ | 0.02194 ** | $LnCO_2$ | 0.03741 ** |
| LnKOF | 0.06654 | LnKOF | 0.07142 * |
| LnGDPpc | 0.02167 ** | LnGDPpc | 0.03611 ** |
| Constant | 1.14748 *** | Constant | 0.93257 *** |

Note: ***, **, and * denote statistical significance at the 1%, 5%, and 10% levels of significance, respectively. Source: Authors' calculations.

**Table 11.** Clusters 2, 4, and 5 statistics.

| Equation | Obs | Parms | RMSE | "R-sq" | Chi | *p*-Value |
|---|---|---|---|---|---|---|
| Cluster 2 | | | | | | |
| LnWGH/LnWGM | 64 | 7 | 0.0153 | 0.9716 | 47.17 | 0 |
| LnWGM/LnWGL | 64 | 7 | 0.0184 | 0.9742 | 44.91 | 0 |
| LnWGWH/LnWGWM | 50 | 7 | 0.0231 | 0.9814 | 36.14 | 0 |
| LnWGWH/LnWGWL | 50 | 7 | 0.0247 | 0.9831 | 48.99 | 0 |
| Cluster 4 | | | | | | |
| LnWGH/LnWGM | 83 | 7 | 0.0124 | 0.9712 | 36.19 | 0 |
| LnWGM/LnWGL | 83 | 7 | 0.0148 | 0.9732 | 47.31 | 0 |
| LnWGWH/LnWGWM | 104 | 7 | 0.0193 | 0.9766 | 51.19 | 0 |
| LnWGWM/LnWGWL | 104 | 7 | 0.0187 | 0.9751 | 50.38 | 0 |
| Cluster 5 | | | | | | |
| LnWGH/LnWGM | 48 | 7 | 0.0138 | 0.9737 | 31.11 | 0 |
| LnWGM/LnWGL | 48 | 7 | 0.0261 | 0.9851 | 37.21 | 0 |
| LnWGWH/LnWGWM | 37 | 7 | 0.0134 | 0.9736 | 42.18 | 0 |
| LnWGWM/LnWGWL | 37 | 7 | 0.0257 | 0.9834 | 48.39 | 0 |

Source: Authors' elaboration.

## 5. Discussion of the Results

As we can see in Tables 7–9, the clusters show very different results, which justifies the cluster approach followed in this paper and also deepens the analysis and results of Nogueira and Madaleno (2023) for the generality of OECD countries.

However, there is a commonality between all clusters: For both the wage gap rates for most workers and the wage gap rates for female workers as a percentage of men's earnings, there is a significant and positive impact between the wage gap rate of workers with full secondary education relative to workers with less than secondary education on the difference between the wage gap rates of workers with higher education and those with only secondary education. As Nogueira and Madaleno (2023) note, this fact may be because workers with intermediate skills can earn higher wages through the accumulation of work experience or vocational training. While this fact does not replace the gains made through education, it has the particularity of influencing the equalization of wages between different groups of workers. While this reality is observed in all clusters, it is particularly striking among workers in countries belonging to cluster 5. Under the ceteris paribus condition, the wage gap between the most qualified groups increases by 0.41% when the wage gap between those with secondary education and those with higher education increases by 1% each. This means that the wage gap for the low-skilled increases the wage gap for higher-skilled professionals, especially in countries where average purchasing power is lower.

It is also confirmed in all clusters that the SBTC approach further widens the wage gap in favor of higher-skilled workers and even contributes to the convergence of women's wage rates as a percentage of men's incomes of the majority of workers. This has been found recently by Messina and Silva (2019), Hutter and Weber (2022), and Nogueira and Madaleno (2023), among others.

One of the great advantages of dividing countries into clusters is the conclusions that can be drawn from the GDP per capita variable. In the cluster consisting of the countries of Northern Europe, the USA, and Australia, this variable helps to reduce wage differentials between all workers. This confirms Kuznets' (1955) hypothesis that an increase in GDP per capita should be accompanied by a reduction in wage differentials, as people with greater purchasing power seek higher skill levels, increasing the number of higher-skilled workers relative to lower-skilled workers and reducing wage differentials on the supply side. In the other two clusters, the GDP per capita variable contributes to widening the wage gap in all the models examined, which may indicate that, in these cases, Kuznets' (1955) claim is not borne out.

In cluster 2, which consists of countries with the highest average GDP per capita, expenditure on education is the variable that contributes most to widening the wage gap between groups of workers, which also contributes to narrowing the wage gap between women's and men's wages. Nogueira and Afonso (2018) also found identical results. In their study, in countries with higher GDP per capita on average, education expenditure is the variable that has the largest impact on the increase in the wage gap. Jacobs and Thuemmel (2022) also reached similar conclusions about the importance of education expenditure in widening the wage gap when considered in relation to academic qualifications.

Similarly, in cluster 2, the impact of trade unions is not seen to be significant in converging or increasing the wage rates of the best-qualified relative to the average-qualified workers, likely because the impact of trade unions is felt more strongly among workers occupying jobs that require lower levels of education (Nogueira and Madaleno 2023). Thus, the impact of trade unions in reducing wage differentials is already evident at the level of lower-skilled workers. In this cluster, globalization contributes to widening the wage gap for lower-skilled workers and also to narrowing the gap between women's and men's wage rates.

Cluster 4 is the one that, on average, has a higher R&D-to-GDP ratio. This fact could mean that the SBTC approach has the largest impact on widening wage gaps in this cluster, especially for workers with higher education compared to those with secondary education. Moreover, for cluster 4 and in the four estimated models, the role of trade unions does not have a significant impact on wage differentials, but on the other hand, carbon emissions contribute with statistical significance to the widening of wage differentials, always benefiting the best qualified. These results do not confirm the claim of Krueger et al. (2021) that workers with increasing qualifications are willing to give up part of their wages for environmental reasons.

As for the Environmental Performance Index, it only contributes positively and with statistical significance to the widening of the wage gap for most workers and for those with tertiary education compared to secondary education. This also does not confirm the findings of Krueger et al. (2021), as it generally does not contribute significantly to wage gaps.

Finally, cluster 5 is the one with lower average GDP per capita, lower average EPI, and higher average union density. As far as union density is concerned, the statistically significant effects are felt in the approach to wage differentials between workers with secondary education and those without, as already demonstrated, for example, by Western and Rosenfeld (2011). Moreover, this is the group where spending on education contributes most to narrowing the wage gap and where there is also a greater contribution to the convergence of women's wages compared to the majority of workers. For example, under the ceteris paribus condition and in the case of female workers, for every 1% increase in education expenditure, the wage gap between those with tertiary education and those with secondary education increases by 0.163%, while for the majority of workers, it increases by only 0.157%.

The limitations of this study are largely related to the time horizon. Although our sample covers thirteen years, the division into clusters leads to a reduction of observations in each cluster, which may change the conclusions to some extent if there were more observations or change the intensity of the coefficients, or even the significance of the variables.

With a higher number of observations per cluster, the results obtained and the variances explained could lead to different conclusions. Another limitation is that there is no data as detailed as that for OECD countries, as the different skill levels in each country have to be taken into account and we, as many other authors, have chosen to conduct this study at the country level. If the sector data were as detailed as those for the countries, the conclusions might be different, as each sector of the economy has its specifications.

## 6. Conclusions and Policy Recommendations

In this innovative paper, we attempt to examine the emergence of wage differentials for most workers in OECD countries, considering the SBTC approach but following the cluster analysis methodology, for the period between 2007 and 2020.

We used the wage gap between university graduates and high school graduates and the wage gap between high school graduates and graduates of schools with low educational attainment. We follow the same criteria for the regressions of women's wage rates as a percentage of men's earnings.

As usual, the R&D expenditure variable is used as a proxy for examining the SBTC approach. We followed this choice and used a wide range of control variables commonly used in this type of work. As already demonstrated by Nogueira and Madaleno (2023), and as we suspect that there are economic reasons that may contribute to the interdependence between the two wage differentials, we used simultaneous equations modeling, and once again, this choice proved to be the right one for all clients.

The first important conclusion that can be drawn, which is only visible through the use of simultaneous equations, is that an increase in the wage gap between workers with and without a high school diploma can lead to an increase in the wage gap between university graduates and high school graduates. This conclusion holds for most workers and the gender wage gap as a percentage of earnings. Given this evidence, we can conclude that there is a convergence of wage rates between the sexes in this way. Through cluster analysis, we can deepen this analysis and mention that there is evidence that in countries with lower GDP per capita values, these effects are amplified. Although workers can increase their wage income through work experience or vocational training, investment in education continues to pay off, with the wage gap widening in favor of better-qualified workers.

As for the SBTC approach, we found that it is significantly present in all clusters and all twelve estimated models. However, it is the cluster with the highest average investment in R&D as a percentage of GDP that achieves the greatest intensity, demonstrating that the benefits accruing from R&D investment feed back to the wages of workers with higher academic qualifications, although they also have a positive impact on the wages of workers with secondary education. This type of investment also contributes to an equalization of wage rates between the sexes. Cluster analysis is used to demonstrate the positive impact of investments in R&D on the wage levels of higher-skilled workers and their contribution to reducing the gender wage gap.

Regarding the union density of workers in OECD countries and its impact on wage differentials, we can find that it has an impact on the level of lower-skilled workers and promotes the alignment of wages of lower-educated workers with those of university graduates, especially in the cluster consisting of countries with lower per capita purchasing power.

The environmental performance index variable, as already demonstrated by Nogueira and Madaleno (2023), has little importance in forming wage differentials and almost always has negative coefficients.

In terms of education expenditure as a percentage of GDP, we find that wage gaps widen in favor of more skilled workers in all clusters, but more so in the cluster consisting of OECD countries with lower GDP per capita. The labor market continues to reward workers who invest in tertiary education, but this is more pronounced in the less affluent OECD countries. On the other hand, the $CO_2$ emissions per capita variable are responsible for widening the wage gap in favor of higher-skilled workers in almost all models, albeit with a residual contribution.

The GDP per capita variable, with its negative coefficients, contributes to a narrowing of the wage gap in all models, being most pronounced in the cluster comprising the OECD countries with the lowest GDP per capita. In this cluster, this variable also contributes most to the convergence of the wage gap between women and men.

Only in the models regressing the wage gap between graduates and non-graduates, the proxy variable of globalization is statistically significant only in clusters 2 and 5. How-

ever, in the Northern European, US, and Australian clusters, it has a larger impact and significantly promotes gender wage rate equalization, positively highlighting the impact of globalization in these regions.

In terms of policy recommendations, we believe that countries should continue to increase spending on education and encourage students to acquire more and more qualifications. This will improve their skills, which will not only increase productivity but also provide workers with better wages, thus benefiting the economy as a whole from spillover effects.

Another important recommendation comes from the results obtained with the SBTC approach. It was evident that R&D investment leads to better wages for workers with tertiary education compared to others. This type of expenditure usually brings benefits to the economy, companies, the nation, and society in general. Because of the improvements achieved and the innovation that accompanies the aforementioned improvements in wage rates, an increase in private or public investment in R&D is therefore desirable. Countries that invest less in R&D can take their cue from the results of the cluster that invests the most and thus increase these investments.

In the future, this individualized study could be extrapolated to other regions of the world, such as African countries or Southeast Asia, through clusters and subdivisions by gender and skill level, to compare the results obtained for OECD countries once we are confronted with other cultural, economic, and social realities.

**Author Contributions:** Conceptualization, M.C.N. and M.M.; investigation, M.C.N.; methodology, M.C.N.; supervision, M.M.; writing—original draft, M.C.N.; writing—review and editing, M.M. All authors have read and agreed to the published version of the manuscript.

**Funding:** This research received no external funding.

**Institutional Review Board Statement:** Not applicable.

**Informed Consent Statement:** Not applicable.

**Data Availability Statement:** Data sharing is not applicable.

**Acknowledgments:** This work was financially supported by the Research Unit on Governance, Competitiveness and Public Policies (UIDB/04058/2020) + (UIDP/04058/2020), funded by national funds through FCT—Fundação para a Ciência e a Tecnologia.

**Conflicts of Interest:** The authors declare no conflict of interest.

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
