# Peer review of "New Evidence about Skill-Biased Technological Change and Gender Wage Inequality"

_economies, doi:10.3390/economies11070193_

Round 1
Reviewer 1 Report
Thanks for letting me review this paper
This study seeks to elucidate the steady increase in the wage gap between more and less skilled workers that has occurred around the world in recent decades, and why this gap is
widening.
Language needs to be checked. It should be ‘new evidence’ rather than ‘new evidences...’.right at the start, and then six lines lower ‘This work intends to deepen the formation of this wage gap’ should say ‘....intends to deepen the understanding of this ....’, ‘bellow’ instead of ‘below’ on so on.
Intro and lit review seem reasonable.
Overall it isn’t clear why using data aggregated to the country level is a valid path of analysis for this research. This would have to be justified in more detail. For instance, industry profiles vary substantially between the countries under investigation, and the analysis does not appear to account for such differences.
In Table 2, what does the ‘average’ column show? Presumably hourly wage, etc? It should specify this.
The separation into the clusters seems debatable. Other clusterings seem possible too, dependent on clustering method. Should be elucidated further.
Regarding tables 7 to 9, explained variances appear extraordinarily, all close to 100 percent, and this while there’s only a small number of observations and predictors.
I suspect there’s significant variance inflation somewhere among the independent variables, and this should be checked and explained.
Tables are difficult to read. Having a different number of columns relating to a) model information and b) coefficients in the same table makes it difficult to understand what’s going on.
Thanks for letting me review this paper
This study seeks to elucidate the steady increase in the wage gap between more and less skilled workers that has occurred around the world in recent decades, and why this gap is
widening.
Language needs to be checked. It should be ‘new evidence’ rather than ‘new evidences...’.right at the start, and then six lines lower ‘This work intends to deepen the formation of this wage gap’ should say ‘....intends to deepen the understanding of this ....’, ‘bellow’ instead of ‘below’ on so on.
Intro and lit review seem reasonable.
Overall it isn’t clear why using data aggregated to the country level is a valid path of analysis for this research. This would have to be justified in more detail. For instance, industry profiles vary substantially between the countries under investigation, and the analysis does not appear to account for such differences.
In Table 2, what does the ‘average’ column show? Presumably hourly wage, etc? It should specify this.
The separation into the clusters seems debatable. Other clusterings seem possible too, dependent on clustering method. Should be elucidated further.
Regarding tables 7 to 9, explained variances appear extraordinarily, all close to 100 percent, and this while there’s only a small number of observations and predictors.
I suspect there’s significant variance inflation somewhere among the independent variables, and this should be checked and explained.
Tables are difficult to read. Having a different number of columns relating to a) model information and b) coefficients in the same table makes it difficult to understand what’s going on.
Author Response
Thanks for letting me review this paper
This study seeks to elucidate the steady increase in the wage gap between more and less skilled workers that has occurred around the world in recent decades, and why this gap is widening.
Language needs to be checked. It should be ‘new evidence’ rather than ‘new evidences...’.right at the start, and then six lines lower ‘This work intends to deepen the formation of this wage gap’ should say ‘....intends to deepen the understanding of this ....’, ‘bellow’ instead of ‘below’ on so on.
Answer:
Thanks for the suggestion. We have changed all occurrences in accordance and we revisited the text in accordance as well doing English editing.
Intro and lit review seem reasonable.
Overall it isn’t clear why using data aggregated to the country level is a valid path of analysis for this research. This would have to be justified in more detail. For instance, industry profiles vary substantially between the countries under investigation, and the analysis does not appear to account for such differences.
Answer:
Thanks for the valuable comment. We completely agree with the reviewer, and it would indeed offer interesting insights considering that different industry profiles among countries would certainly lead to interesting results. However, usually, studies considering the same thematics are practically all built considering countries and not industries once there is not enough data. Which is also the case in our article. We would like to do that, maybe in the future, and as such we left this as a limitation and future research tip.
In Table 2, what does the ‘average’ column show? Presumably hourly wage, etc? It should specify this.
Answer:
Apologies for the missing. The text was corrected in accordance.
The separation into the clusters seems debatable. Other clusterings seem possible too, dependent on clustering method. Should be elucidated further.
Answer:
Thanks for the valuable suggestion. We are aware that some other methods and distances would probably offer different results. However, we followed the most used ones as suggested by other authors in the literature and we left two references that argue in favor of the method we have used in this article as well. Therefore, we believe, following the existing literature, that this is the most suitable agglomeration method.
Regarding tables 7 to 9, explained variances appear extraordinarily, all close to 100 percent, and this while there’s only a small number of observations and predictors.
Answer:
Thanks for the suggestion. We agree with the reviewer in the sense that the reduced number of data for each variable/country might have influenced the explained variables. As a limitation we have already suggested the collection of more data to check if results would be kept.
I suspect there’s significant variance inflation somewhere among the independent variables, and this should be checked and explained.
Answer:
Thanks for the valuable suggestion. To infer variance inflation, results were computed, presented, and explained within the text. Table 4 contains this new information and results point to values lower than 2 in all variables.
Tables are difficult to read. Having a different number of columns relating to a) model information and b) coefficients in the same table makes it difficult to understand what’s going on.
Answer:
Thanks for the valuable suggestion. We tried to simplify in order not to increase the number of tables, but considering the reviewer's comment we have provided more table results to turn easier the analysis.
Reviewer 2 Report
The authors proposed an interesting study, “New evidences about Skill-Biased Technological Change and Gender Wage Inequality.” The paper is well-structured and conveys a deal of information. I want to suggest a few suggestions to improve the manuscript's quality and readability.
1. If the authors use abbreviations, they must be used in a systematic way (e.g., OECD, R&D).
2. There is a need to do a more rigorous and systematic literature review. The authors should clearly mention the literature gap.
3. The theoretical explanation section is missing. Kindly add a theoretical part with concrete and cohesive arguments regarding the studied variables.
4. Limitations and future research directions are missing.
5. Lastly, there are many short and meaningless sentences in the manuscript, which can affect the quality of the manuscript. Please revise carefully.
Author Response
Reviewer 2
The authors proposed an interesting study, “New evidences about Skill-Biased Technological Change and Gender Wage Inequality.” The paper is well-structured and conveys a deal of information. I want to suggest a few suggestions to improve the manuscript's quality and readability.
- If the authors use abbreviations, they must be used in a systematic way (e.g., OECD, R&D).
Answer:
Thanks for the valuable suggestion. All abbreviations have been revised in accordance.
- There is a need to do a more rigorous and systematic literature review. The authors should clearly mention the literature gap.
Answer:
Thanks. The literature review has been improved following suggestions 2 and 3. We have reinforced the contribution of the article, clearly identifying the gap in the introduction section.
- The theoretical explanation section is missing. Kindly add a theoretical part with concrete and cohesive arguments regarding the studied variables.
Answer:
Following the previous and this suggestion, we have offered now a clear explanation of the theoretical arguments justifying the use of these explanatory variables. We suggest a new reading of the article section literature review.
- Limitations and future research directions are missing.
Answer:
Thanks for the suggestion. Since the other reviewer also asked to provide more future research directions and considering that your and others' comments have offered new limitations and future research directions, these have been reinforced in the conclusions section.
- Lastly, there are many short and meaningless sentences in the manuscript, which can affect the quality of the manuscript. Please revise carefully.
Answer:
Apologies for the missings and errors. A new English editing has been done during the revision of the article and we hope to have conformed to all the required suggestions.
Round 2
Reviewer 1 Report
The revision has addressed the main points of my previous review and the paper is now significantly improved. I support the publication of this paper.